

# Probabilistic Seismic Landslide Hazard Assessment Considering Different Scenarios of Earthquake and Rainfalls in Bomi, China

Chen Shuai[1, 2], Miao Zelang[2, 3], Wu Lixin[2, 3*]

[1]College of Advanced Interdisciplinary Studies, Central South University of Forestry and Technology, Changsha 410004, Hunan, China
[2]Laboratory of GeoHazards Perception, Cognition and Predication, Central South University, Changsha 410083, Hunan, China
[3]School of Geoscience and Info-physics, Central South University, Changsha 410083, Hunan, China

*Correspondence to*: Wu Lixin (wulx66@csu.edu.cn)

**Abstract:** Probabilistic seismic landslide hazard assessments are critical to infrastructure safety and disaster mitigation in earthquake-prone zones. Previous studies on the probabilistic seismic landslide hazard (PSLH) assessment considered only earthquakes, while rainfall was rarely or not yet considered, which might affect significantly the spatio-temporal pattern of potential seismic landslides. Considering the uncertain features of both earthquake and rainfalls, we developed a novel method for PSLH assessment referring to static factors (geology, topography, and landuse/landcover) and dynamic factors (earthquake and rainfalls), and assessed the PSLH in Bomi, China, which is a strong earthquake-prone zone threatened by heavy rainfalls in the southeast of the Tibet plateau. Firstly, the earthquake parameters under four kinds of earthquake scenarios, being frequent, occasional, rare, and extremely rare, were obtained with the probabilistic seismic hazard analysis method to quantify the effect of future earthquakes. Secondly, we quantified the spatio-temporal distribution of the soil slope saturation with a rainfall infiltration model considering the monthly different rainfalls. Then, considering the different scenarios of both earthquake and rainfall, we assessed in detail the PSLH with a permanent displacement model. The results show that the risky zones of seismic landslide hazards differ significantly in Bomi under different scenarios, where high and extremely high hazard zones concentrate mainly in the south part; and the pattern of seismic landslide hazards changes a lot with monthly differential rainfalls. The method presented in this study is meaningful for the prevention and mitigation of seismic landslides in other mountainous areas threatened by strong earthquakes and suffering from heavy rainfalls.

**Keywords:** Earthquake; seismic landslide; rainfall; probabilistic seismic landslide hazard; assessment.

## 1. Introduction

Seismic landslides are hazard phenomena of slope stability-losing induced directly or indirectly by earthquakes, which result in significant casualties, infrastructure damage, and substantial property losses as a secondary geological disaster(Fan et al., 2019; Tang et al., 2017). For example, the 2008 Mw 7.9 earthquake occurred in Wenchuan induced more than 197,000 seismic landslides, resulting in the death of 69227 people(Cui et al., 2011; Xu et al., 2014). Probabilistic Seismic Landslide



Hazard (PSLH) assessment combines the probabilistic seismic hazard assessment and the seismic landslide hazard assessment to assess the seismic landslide hazards under future earthquake scenarios. The results of the PSLH assessment can provide important references for medium- and long-term landuse planning of urbanization and engineering site selection of major infrastructures(Gerstenberger et al., 2020; Rollo and Rampello, 2021; Wang and Rathje, 2015).

Data-driven analysis (DDA) and permanent displacement model (PD model) are two typical methods usually used in seismic landslide hazard assessment(Fan et al., 2019). The DDA method establishes a mathematical model based on the relationships between landslides and complex geological environmental factors, which are derived from the historical seismic landslide inventory, to assess the seismic landslide hazard(Bloom et al., 2023; Shao and Xu, 2022). Its effectiveness depends on the quality of the seismic landslide inventory including completeness(Chen et al., 2021; Chen et al., 2020; Du et al., 2020). The

DDA method has been mainly used in the inversion studies of historical seismic landslide hazards recently. Different from the DDA model, the PD model does not require a landslide inventory and can assess the seismic landslide hazard more reliably by considering comprehensively the formation mechanism of seismic landslides(Delgado et al., 2020; Jibson, 2011). Rollo (2021) applied successfully the PD model to assess potential seismic landslide hazards for the zone of high seismic activity in Irpinia, southern Italy and analyzed the spatial distribution of potential seismic landslide hazards under different

future earthquake scenarios. Hence, the PD model is of potentiality supporting the PSLH assessment in this study.

Earthquakes and rainfalls are critical factors that affected the distribution of seismic landslides(Gerstenberger et al., 2020; Havenith et al., 2022; Rollo and Rampello, 2021). Previous studies showed that the number and size of seismic landslides increased significantly due to the combined effect of the earthquake and rainfall. For example, Sassa (2007) revealed that the number and scale of seismic landslides increased significantly in the zone suffered from heavy rainfall before the

earthquakes, after analyzing carefully the seismic landslides induced by the 1995 southern Hyogo Prefecture Mw 7.2 earthquake and the 2004 central Niigata Prefecture Mw 6.8 earthquake. Pu (2021) investigated the triggering mechanism of landslides under the combined effect of earthquake and rainfall through indoor geotechnical experiments, slope simulation experiments, and numerical simulations. The rainfall infiltration was proved affecting significantly the slope stability and making the slopes more susceptible to the formation of landslides if triggered by an earthquake. Although both earthquake

and rainfall can affect the distribution of seismic landslide hazards(Chen et al., 2020; Gnyawali et al., 2020), few studies have considered the combined effect of earthquake and rainfall in the process of PSLH assessment.

This paper aims to develop a novel PSLH assessment method considering both earthquakes and rainfall. With Bomi, China, a particular region of high seismic activity located in the southeast part of Tibet, being a case study, the developed method was demonstrated under different earthquake scenarios referring to the monthly different rainfalls. The spatio-temporal

pattern of potential seismic landslide hazards threatened by future earthquakes and rainfalls in Bomi is revealed, which provides useful information for local social-economic development as well as method reference for mitigating the seismic landslides in other earthquake-prone mountainous zones suffering from heavy rainfalls.



## 2. Methodology

### 2.1 Permanent displacement (PD) model

The PD model was first proposed by Newmark in 1965 for the stability analysis of dam suffering from an earthquake(Newmark, 1965), which has become a widely used method in seismic landslide hazard assessment in recent years(Jibson, 1993; Wieczorek et al., 1985). The strength parameters of the slopes, being the primary input of the PD model, are critical for seismic landslide hazard assessment(Dreyfus et al., 2013). Affected by the complex geo-environment, accurate strength parameters of the slopes are hard to obtain; while the simplified empirical strength parameters were usually

adopted ignoring the spatial heterogeneity of regional rock mass, which leads to great uncertainty in seismic landslide hazard assessment in the condition of complex geo-environments (Gallen et al., 2015; Shinoda et al., 2019). An improved PD method considering the spatial heterogeneity of the strength parameters of the slopes, coined as CRMSH-PD, was developed recently, which has been applied effectively in the seismic landslide hazard assessment of the 2008 Wenchuan M7.9 earthquake and the 2013 Ludian M7.3earthquake in China(Chen et al., 2023). Based on CRMSH-PD, we developed the

PSLH assessment model and took Bomi for a case study, in that the geo-environment characteristics of Bomi are much similar to that of Wenchuan and Ludian, which located also in the east part of Tibet.

Different from the classical PD model, the yield acceleration ($K_{ym}$) in the CRMSH-PD model is expressed as:

$$K_{ym} = f_M \times K_y \tag{1}$$

where, $f_M$ is an indicator quantifying the strength spatial heterogeneity of the slopes in complex geo-environments, $K_y$ is

the initial yield acceleration in the classical PD model.

The strength spatial heterogeneity of the slopes in complex geo-environments can be calculated with an empirical equation (Chen et al., 2023), as follows:

$$f_M = \frac{1}{1+e^{0.20 \times f_{cur} + 0.23 \times f_{fault} + 0.13 \times f_{river} - 0.51 \times f_{relief}}} \tag{2}$$

where, $f_{cur}$, $f_{fault}$, $f_{river}$ and $f_{relief}$ are topography curvature, distance to the fault, distance to the river, and topography relief,

respectively.

The initial yield acceleration can be calculated as:

$$K_y = \left\{ \frac{(c')}{\gamma t \sin a} + \frac{\tan(\varphi')}{\tan a} - \frac{m\gamma_w \tan(\varphi')}{\gamma \tan a} - 1 \right\} \sin a \tag{3}$$

where, $c'$ and $\varphi'$ are the effective cohesive and effective friction of the rock mass, respectively; $\gamma$ and $\gamma_w$ are the unit weight of the rock mass and the groundwater; $t$ is the depth of the potential landslide; $a$ is the angle of the sliding surface;

$m$ is the saturation of the potential landslide.



Permanent seismic displacement is an effective indicator reflecting the seismic landslide hazard, which can be calculated with the empirical displacement model based on different ground motion parameters, such as peak ground acceleration (PGA) and Arias intensity. This study employed the empirical PGA-based displacement model (Jibson, 2011) to calculate the permanent seismic displacement as:

$$\log(D_N) = 0.215 + \log\left[\left(1 - \frac{K_{yM}}{PGA}\right)^{2.341} \times \left(\frac{K_{yM}}{PGA}\right)^{-1.438}\right] \tag{4}$$

where, $D_N$ is the permanent seismic displacement in terms of cm; PGA and $K_{yM}$ are in terms of $g$.

## 2.2 Rainfall infiltration model

Rainfall infiltration is an important factor influencing the saturation of soil slopes, and the Green-Ampt model (Green and Ampt, 1911) is a classical rainfall infiltration model used to quantify the saturation of soil slopes and has been widely used in slope stability analysis for its easy calculation and clear physical meaning(Guzzetti et al., 2020; Lombardo et al., 2020). This study adopted the Green-Ampt model to simulate rainfall infiltration and quantify soil slope saturation.

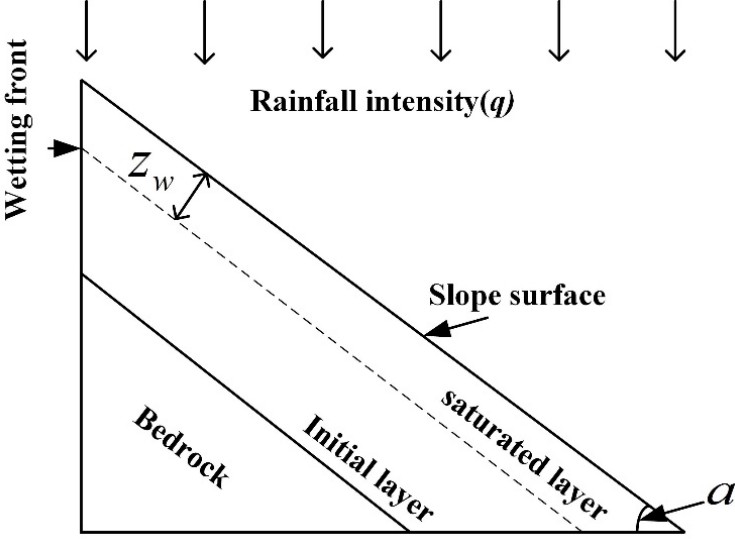

**Figure 1 Simplified schematic diagram of the Green-Ampt model.**

Figure 1 shows a simplified schematic of the Green-Ampt model, in which a wetting front is assumed to be present within the soil slope under rainfall. The wetting front separates conceptually the slope into an upper saturated layer and a low initial layer (unsaturated layer). Based on Darcy's law, the cumulative infiltration capacity of time ($I$) can be calculated as:

$$I = (\theta_s - \theta_i)z_w \tag{5}$$



where, $\theta_s$ and $\theta_i$ are the water content of the saturated layer and the initial layer, respectively; $z_w$ is the wetting front depth.

The infiltration rate ($i$) can be calculated with the time-derivation of $I$ as:

$$i = \frac{dI}{dt} = (\theta_s - \theta_i)\frac{dz_w}{dt} \tag{6}$$

where, $t$ is the rainfall duration.

The wetting front depth can be calculated under different rainfall intensities. In case of rainfall intensity ($q$, m/s) being less than the saturated hydraulic conductivity ($K_s$, m/s), the wetting front depth can be calculated as follows:

$$z_w = \frac{q\cos a}{(\theta_s - \theta_i)}t \tag{7}$$

where, $a$ is the slope gradient (°).

In case of rainfall intensity being greater than the saturated hydraulic conductivity, the wetting front depth can be calculated differently considering different durations of rainfall as follows:

$$z_w = \begin{cases} \dfrac{q\cos a}{(\theta_s - \theta_i)}t, 0 < t <= t_p \\ \dfrac{K_s}{(\theta_s - \theta_i)}\dfrac{z_w\cos a + s_f}{z_w}t, t > t_p \end{cases} \tag{8}$$

where, the $t_p$ is the time required for the rainfall to infiltrate to the slope sufficiently making the wetting front saturated, which can be obtained as:

$$t_p = K_s \frac{S_f(\theta_s - \theta_i)}{q^2(\cos a)^2 - qK_s(\cos a)^2} \tag{9}$$

It should be noted that the value of wetting front depth depends on the rainfall intensity and the duration of rainfall. Previous studies suggested that the distribution of landslides is highly related to monthly rainfalls (Mathew et al., 2014; Tang et al., 2017). Therefore, we selected the monthly cumulative rainfall intensity in this study as the input of the Green-Ampt model.

**2.3 Seismic landslide hazard assessment incorporating rainfalls**

The saturation degree of soil slopes is a critical factor affecting the spatial distribution of the landslide hazard, which is highly correlated with the rainfalls. To incorporate the rainfall effect on seismic landslide hazard assessment, this study employs the saturation degree of soil slopes to the CRMSH-PD model, which is an improved PD model for seismic landslide hazard assessment. The saturation of the soil slopes can be calculated using the following equation:

$$m = \min(\frac{z_w}{t}, 1) \tag{10}$$





where, $m$ is the saturation of the slopes, $t$ is the depth of the potential landslide; $z_w$ is the wetting front depth, which can be calculated with the Green & Ampt model referring to the rainfall information. The calculated saturation of the soil slopes is then used in the CRMSH-PD model as in Eq. (3), and the seismic landslide hazard is then assessed. Fig.2 shows the flowchart of the seismic landslide hazard assessment incorporating rainfalls.

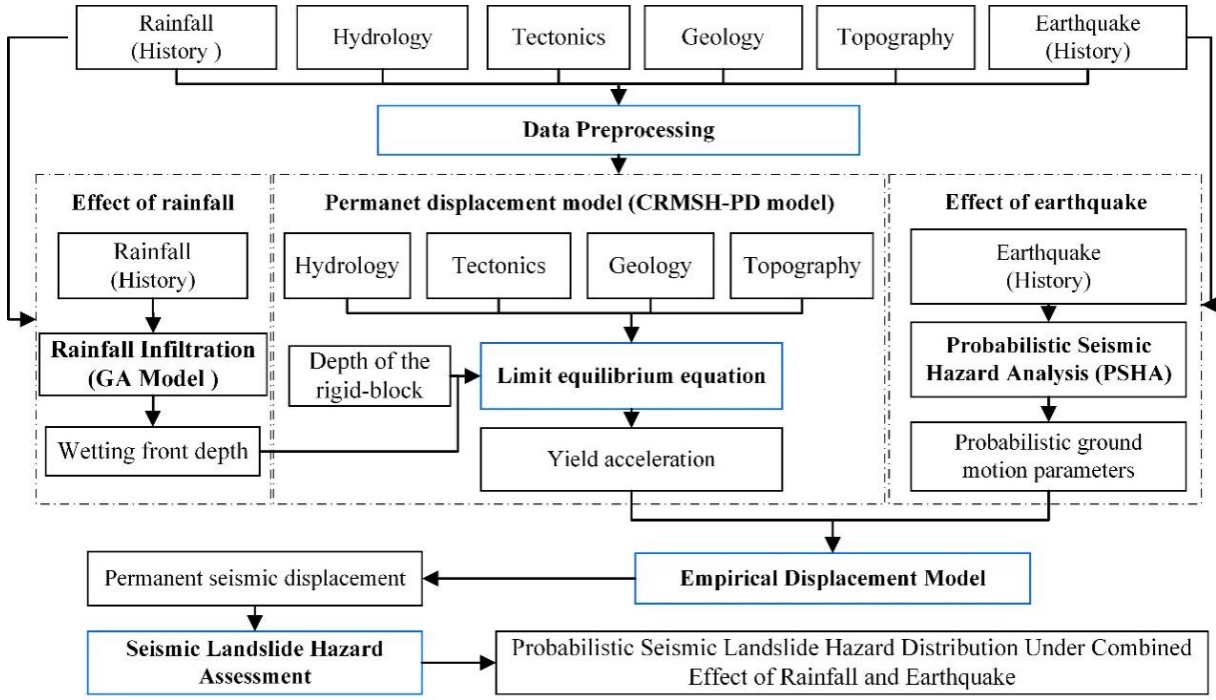

**Figure 2 Flowchart of the novel method for PSLH assessment considering both earthquakes and rainfalls.**

## 3 Case study and data sets

### 3.1 Study area

Bomi is located in the southeast of the Tibetan, covering an area of 16,700 km² about, and shows complicated topography conditions, with a maximum, minimum, and average elevation of 6,569m, 1,998m, and 4,391m, respectively. The rainfall pattern in Bomi varies significantly in both spatial and temporal domains, with an annual average of nearly 900 mm, mainly concentrated in June, July, and August.

Seismicity is very active in Bomi and its nearby area, and strong earthquakes have occurred frequently because of the regional strong tectonic movements(Zhao et al., 2023). Fig. 3 shows the historical earthquakes in Bomi and its nearby (within a 100km buffer zone) from 1900 to 2024. The M8.6 Motuo earthquake in 1950 was one of the strongest earthquakes since 1850, which induced a large number of collapses, landslides, and other geologic hazards that caused a large number of casualties(Li et al., 2021). As a result, the scale and frequency of geological hazards in Bomi increased significantly




compared to that before the great earthquake. For example, on 9 April 2000, a catastrophic landslide occurred in Yigong village, Bomi, and formed a dam with a total accumulated volume of approximately $3.00 \times 10^8$ m$^3$; the dam broke on 10 June

2000 and initiated a huge flooding event, resulting in extensive loss of human life and great damage to the property along the riverway of Yalung Zangbo River in China and India(Gao et al., 2023).

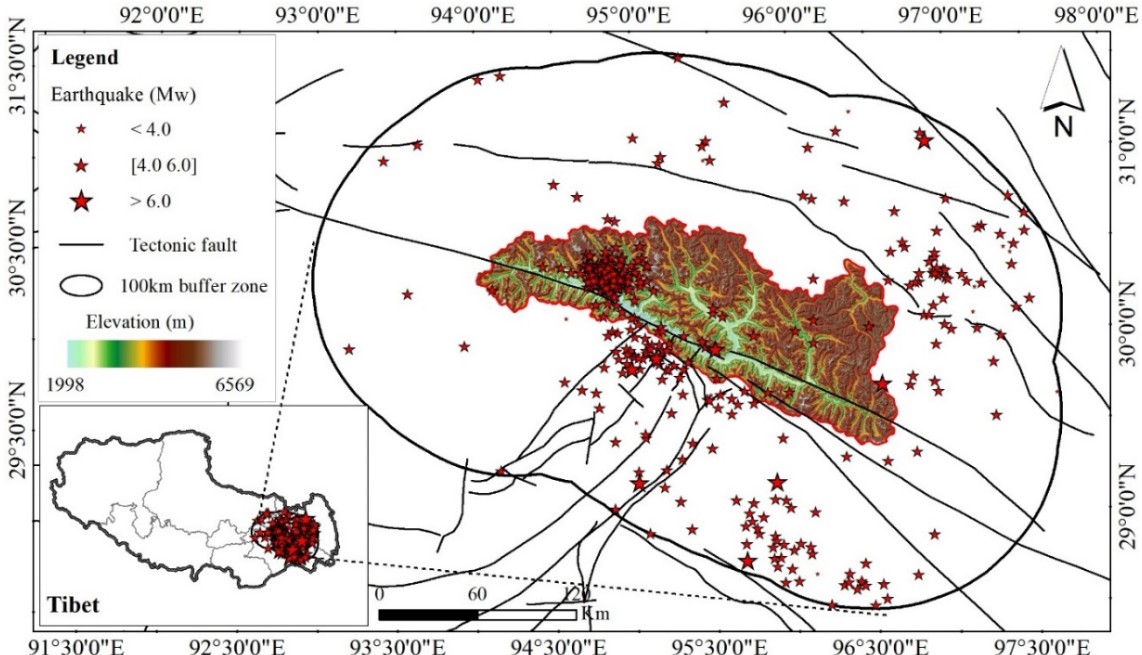

**Figure 3 Geographic location of Bomi and the distribution of tectonic faults and historical earthquakes.**

## 3.2 Data sets

This study considers various environmental factors associated with seismic landslides, which were summarized as static factors and uncertain dynamic factors. All factors are converted into raster data with the same resolution as that of SRTM DEM (30 × 30 m) and re-projected to a uniform spatial coordinate system (UTM-zone 46, WGS 84).

### 3.2.1 Static factors

Static factors mainly include geology, topography, and soil slope that do not change significantly over time. Geology is a key

factor affecting the distribution of landslides. This paper selects the physical and mechanical properties of slopes as the main geologic factors since they are key input parameters for the CRMSH-PD and Green-Ampt models. According to the 1:250,000 engineering geological map and previous studies(Chen et al., 2019; Ma and Xu, 2019), the slopes in Bomi were divided into five different groups, i.e., Groups 1~5. Fig. 4a shows the distribution of different Groups in Bomi. The mechanical parameters (unit weight, effective friction angle, and effective cohesion, as in Table 1) of these Groups are then



assigned referring to previous studies(Du et al., 2022). The mean mechanical parameters of different Groups were used in the CRMSH-PD model in this study.



**Figure 4 Maps of geo-environmental factors used for the PSLH assessment in Bomi, China.**

Physical properties, such as the saturated water content, the initial water content, and the saturated hydraulic conductivity,

are also important in landslide hazard assessment. Since these parameters are not yet available at a regional scale, we assume





that all the Groups in Bomi have the same saturated water content, the initial water content, and the saturated hydraulic conductivity. According to the previous studies in Bomi, the saturated water content of the slopes is usually in the range of 30-50%, and the initial water content is in the range of 10-30%(Liu et al., 2020; SU and LI, 2020). We adopt the average values in previous studies as the saturated water content and initial content of the slopes, being 40% and 20%, respectively.

**Table 1 The mechanical strength parameters of different geological groups in Bomi**

| Group | Main lithology | $c'$(kPa) | $\varphi'$(°) | $\gamma$ (kN/m³) |
|-------|----------------|-----------|---------------|------------------|
| Group 5 | Moraine. | 29-35 | 34-40 | 20-24 |
| Group 4 | Granite, granodiorite, basalt, etc. | 27-33 | 34-40 | 18-22 |
| Group 3 | sandstone, calcareous slate, quartzite, etc. | 24-30 | 32-38 | 16-20 |
| Group 2 | Mudstone, shale, slate and graywacke interbedded, feldspathic quartz sandstone, etc. | 22-28 | 29-35 | 14-18 |
| Group 1 | Quaternary loose accumulation. | 17-23 | 22-28 | 10-14 |

The topography is another static factor affecting the distribution of the landslides. Retrieving from the SRTM DEM data with the spatial analysis tools in ArcGIS, we obtained the topography slope, topography curvature, and topography relief factors for PSLH assessment as in Fig. 4b, c, and d. The distance to the fault was selected to reflect the effect of tectonic distribution, which was extracted from the 1: 2,500,000 fault distribution map using ArcGIS, and was further divided into 10

classes as in Fig. 4e. The distance to the river was selected to reflect the effect of the hydrography distribution on landslide, which was extracted from the 1:1,000,000 river distribution using ArcGIS, and was further divided into 7 classes as in Fig. 4f.

The landuse\landcover data were used to extract the soil slopes in this study(Zhang et al., 2024). Fig. 4g shows the most recent distribution of the landuse\landcover in Bomi. Considering that vegetation mostly covers the area of soil ground and

slopes, the regions of cropland, forest shrub, and grassland were assumed as soil ground or slopes in this study. The results indicated that the majority of the study area belongs to the soil slopes, which account for 63% of the total area. In addition, according to the historical landslide inventory (shown in Fig. 4h), more than 90% of the historical hazards were distributed in the region of the soil slopes, meaning that the soil slopes are more susceptible to landslide hazards.

### 3.2.2 Uncertain dynamic factors

Uncertain dynamic factors are those behaving uncertainly and might change significantly over time, such as earthquakes and rainfalls. The spatio-temporal distribution of rainfall in Bomi was derived from the historical rainfall records. Fig.5 shows the spatial distribution of averaged monthly cumulative rainfall in Bomi during 2000-2021. The average monthly cumulative rainfall in Bomi has a maximum of 83.62 mm and a minimum of 49.74 mm, with also significant differences in spatial distribution. Generally, heavy rainfalls are distributed mainly in the region of low elevation, such as that along the Parlung

Zangbo River, and low rainfalls are concentrated in the north, south, and east parts of Bomi. The distribution of the rainfall



in Bomi varied significantly both in spatial and temporal domains. As shown in Fig. 6, the rainfalls in Bomi concentrated mainly from June to August, with a monthly average of 198 mm, while in January, February, November, and December, the monthly average of 10 mm was much lower.

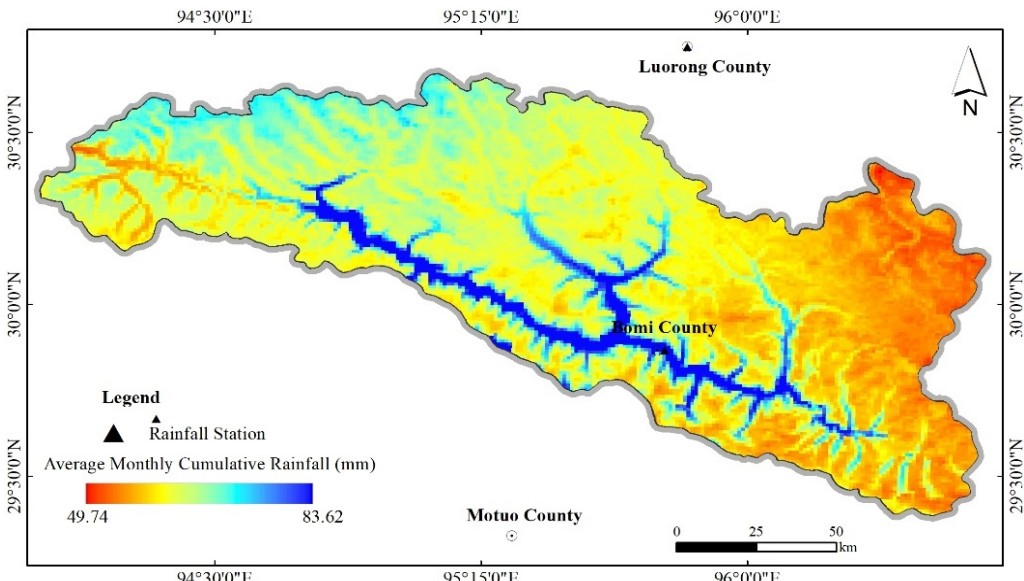

**Figure 5 Map of averaged monthly cumulative rainfall in Bomi 2000-2021.**





**Figure 6 Maps of average rainfall in different months in Bomi 2000-2021.**



Seismicity or PGA (the peak ground acceleration parameter resulted from an earthquake) is another dynamic factor of great
uncertainty, with different spatial distribution characteristics under different seismic scenarios. In this study, the future
earthquake is categorized into different scenarios(General Administration of Quality Supervision et al., 2015): frequent (with
a probability 63% in 50 years), occasional (with a probability 10% in 50 years), rare (with a probability 2% in 50 years), and
extremely rare earthquake occurrence (with a probability 0.5% in 50 years), considering different exceedance probability
conditions. The PGA was derived from the fifth generation of the national ground motion parameter zoning map released in
2015. As shown in Fig.7, the corresponding PGA map was generated using the probabilistic seismic hazard assessment
method referring to different earthquake scenarios. The areas of high PGA values are distributed mainly in the south, while
decreasing from south to north indicating low seismic impact on the north part of Bomi.

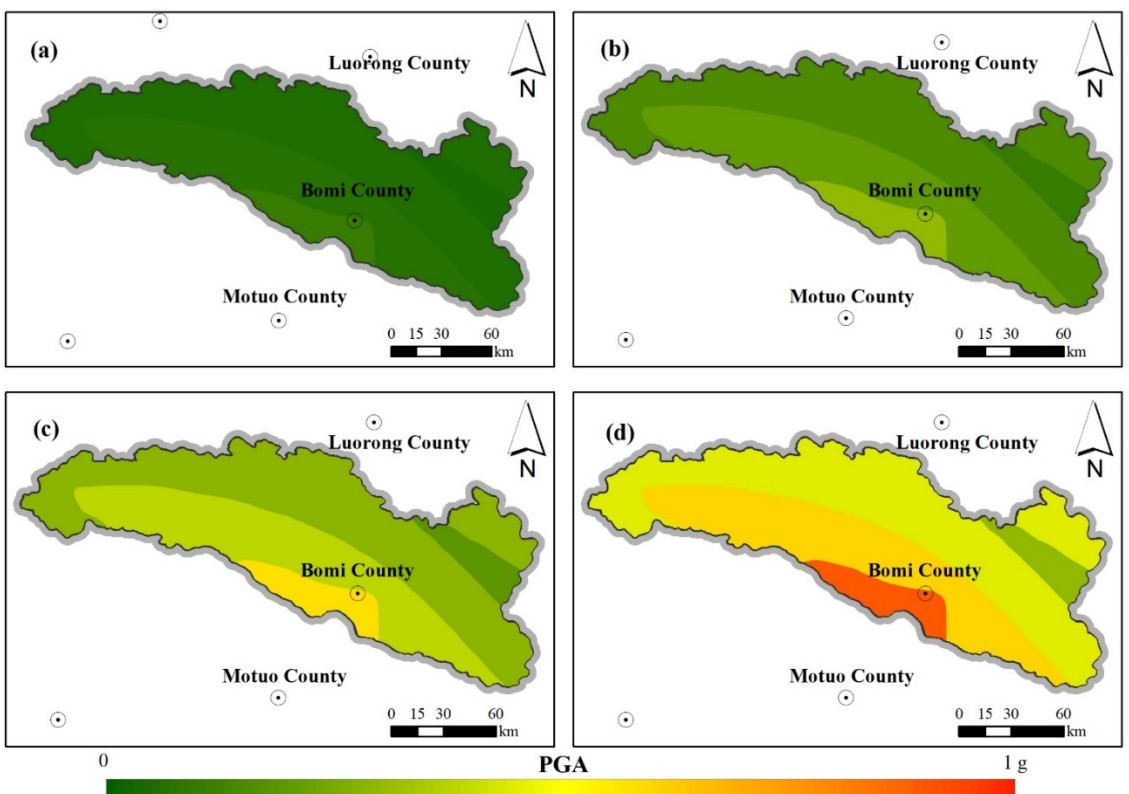

**Figure 7 PGA maps of Bomi in the condition of frequent (a), occasional (b), rare (c), and extremely rare (d) earthquake occurrence**
**scenarios.**





## 4. PSLH assessment

Considering the combined effect of earthquake and rainfall on seismic landslides in Bomi and aiming to make a contrast analysis, this study assessed the PSLH for different earthquake scenarios with a consideration of rainfall or not. A total of 20 scenarios, including 4 without rainfall and 16 with rainfall, were made for PSLH assessment (see Table 2).

**Table 2 Twenty scenarios designed for the seismic landslide hazard assessment in Bomi, China**

| Scenarios | Rainfall conditions | Earthquake scenarios |
| --- | --- | --- |
| 1 | | Frequent earthquake occurrence |
| 2 | Ignored | Occasional earthquake occurrence |
| 3 | | Rare earthquake occurrence |
| 4 | | Extremely rare earthquake occurrence |
| 5 | | Frequent earthquake occurrence |
| 6 | Averaged monthly cumulative | Occasional earthquake occurrence |
| 7 | rainfall in 2000-2021 | Rare earthquake occurrence |
| 8 | | Extremely rare earthquake occurrence |
| 9 | Average rainfall in January | Occasional earthquake occurrence |
| 10 | Average rainfall in February | Occasional earthquake occurrence |
| 11 | Average rainfall in March | Occasional earthquake occurrence |
| 12 | Average rainfall in April | Occasional earthquake occurrence |
| 13 | Average rainfall in May | Occasional earthquake occurrence |
| 14 | Average rainfall in June | Occasional earthquake occurrence |
| 15 | Average rainfall in July | Occasional earthquake occurrence |
| 16 | Average rainfall in August | Occasional earthquake occurrence |
| 17 | Average rainfall in September | Occasional earthquake occurrence |
| 18 | Average rainfall in October | Occasional earthquake occurrence |
| 19 | Average rainfall in November | Occasional earthquake occurrence |
| 20 | Average rainfall in December | Occasional earthquake occurrence |

### 4.1 Considering different earthquake scenarios

Firstly, the rainfall was ignored to quantify the distribution of seismic landslide hazards under different earthquake scenarios, and the saturation of the slopes was set as 0. Since the global empirical depth parameter of the potential landslide was usually used for the CRMSH-PD model in previous studies(Shinoda and Miyata, 2017; Zang et al., 2020), we set the depth of potential landslides as 3 m referring to the depth of landslides from the historic landslide inventory in Bomi. According to the yield acceleration calculated with the CRMSH-PD model, the study area is divided into five susceptibility levels. Higher susceptibility levels correspond to lower slope stability, and more historical landslides developed. The results show that the history landslides are distributed mainly in areas with high and extremely high susceptibility levels, accounting for 76.3% of the total landslides, and the number of landslides increases significantly with the landslide susceptibility level. The yield acceleration calculated based on the CRMSH-PD model can reflect the slope stability to a certain extent, which is applicable to the subsequent PSLH assessment.





**Figure 8 Maps of seismic landslide hazard levels in conditions of frequent (a), occasional (b), rare (c), and extremely rare (d) earthquake occurrence scenarios.**



Then, the permanent seismic displacements of the study area under different earthquake scenarios were calculated with the CRMSH model, and further divided into 4 levels: low (<1 cm), moderate (1-5 cm), high (5-15 cm), and extremely high hazard level (>15 cm), referring to the USGS division criteria(Jibson and Michael, 2009). Fig. 8 shows the distribution of seismic landslide hazards in the condition of four possible earthquake scenarios. It shows that the north part of Bomi, as well as the vicinity of Bomi town, behave at a higher seismic landslide hazard level than other areas, in all scenarios. Particularly,

for the scenarios of frequent and occasional earthquake occurrence, only a small area is assessed as high hazard level, which is distributed mainly in the south of Bomi; for the scenarios of rare and extremely rare earthquake occurrence, the area of high and extremely high levels increases significantly, which scatters around the Bomi town and along the Parlung Zangbo River.

Statistics show that the areas of moderate, high, and extremely high PSLH levels increase significantly with the earthquake

scenario changing from frequent occurrence to extremely rare occurrence. For the frequent occurrence, an area of only 422.74 km$^2$ is assessed as an extremely high seismic landslide hazard level; while the area enlarges to 734.41, 908.47, and 1,172.10 km$^2$ for the occasional, rare, and extremely rare occurrence, respectively, with an enlargement factor being 1.74, 2.15, and 2.77, respectively. It indicates that a large area in Bomi will be in high and extremely high seismic landslide hazard levels if Bomi suffers from strong earthquakes in the future.

The differences in the seismic landslide hazard assessment under the four earthquake scenarios are shown in Fig. 9. Insignificant difference in the spatial distribution of PSLH was observed under the scenarios of frequent and occasional earthquake occurrence (Fig. 9a), where most areas belong to low hazard level. However, if Bomi suffers from strong earthquakes, i.e., under the scenarios of the rare or extremely rare occurrence, the PSLH level increased significantly by 1 to 2, or even 3 levels, as compared to that of occasional occurrence (Fig. 9b and Fig. 9c). The areas of high PSLH level

distribute mainly in the south part of Bomi. It is worth noting that the south part of Bomi, which is the critical corridor of the G317 national road and the Yaan-Linzhi railway under construction, is densely populated and embodies important infrastructures. Great attention and precautionary measures are necessary for disaster prevention in Bomi.



**Figure 9 Changes of PSLH level in the condition of four earthquake occurrence scenarios: (a) Fig. 8a minus Fig. 8b, (b) Fig. 8c minus Fig. 8b, (c) Fig. 8d minus Fig. 8b.**






## 4.2 Considering also monthly rainfalls

To simplify the operation and reduce the input parameters, we assumed that the saturated hydraulic conductivity is greater than the rainfall in Bomi, and thus the wetting front depth can be calculated with the Green-Ampt model.

### 4.2.1 Impact of rainfall's spatial variability

The averaged monthly cumulative rainfall derived from the 2000-2021 rainfall data (Fig. 5) was used to quantify the additional impact of rainfall's spatial variability on PSLH. As shown in Fig.10, the area of extremely high hazard zones gets enlarged to 484.04, 822.27, 1024.50, and 1329.60 km² in the frequent, occasional, rare, and extremely rare earthquake occurrence scenarios, with an enlargement factor being 1.15, 1.12, 1.13, and 1.13 times, respectively, as compared to that rainfall was ignored. Particularly, the majority of areas behaving heavy rainfall in Bomi are assessed as high or extremely

high PSLH levels if rare or extremely rare earthquakes occur in the future. The results indicate that the spatial variability of rainfall is able to affect significantly the PSLH as well as its spatial distribution.

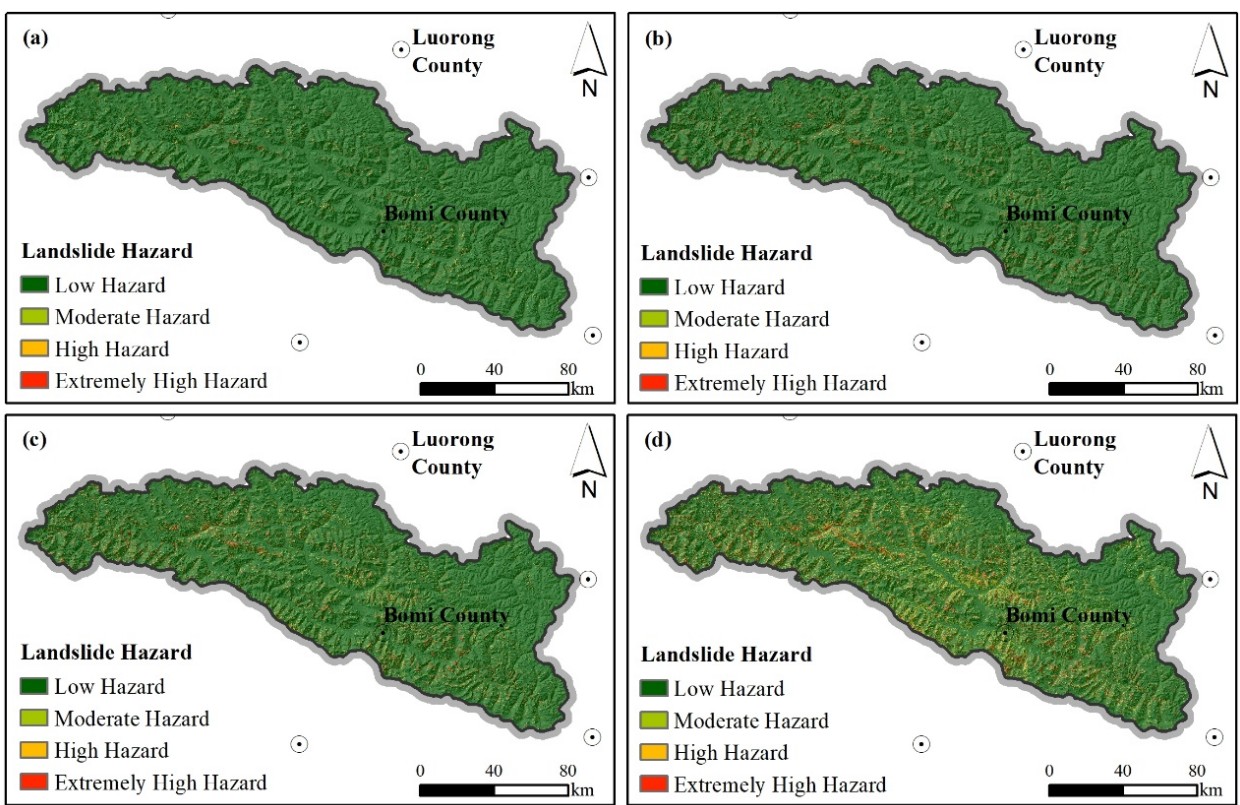

**Figure 10 PSLH maps considering the spatial variability of rainfall in conditions of frequent (a), occasional (b), rare (c), and extremely rare (d) earthquake occurrence scenarios.**



### 4.2.2 Impact of rainfall's monthly difference

Figure 11 shows the produced maps of PSLH in Bomi in different months referring to the different monthly averages of rainfalls. It shows that there are significant differences in the spatio-temporal pattern of PSLH between and among different months, even in the same earthquake scenario. Specifically, in August which has heavy rainfall, the area of high and extremely high PSLH levels is the largest, while it decreases significantly in January, February, November, and December, as a result of much less rainfall. It should be noted that these differences get more significant in conditions of rare and extremely rare earthquake scenarios.





**Figure 11  PSLH maps in different months referring to different earthquake occurrence scenarios.**




The change of PSLH area in different earthquake occurrence scenarios is shown in Fig. 12. For the four earthquake
occurrence scenarios, the temporal pattern of moderate, high, and extremely high PSLH levels is correlated with monthly
rainfall. It suggests that in response to future earthquakes, various prevention and control measures should be taken in
different months to better mitigate the effects of potential seismic landslide hazards affected by the changing rainfall.

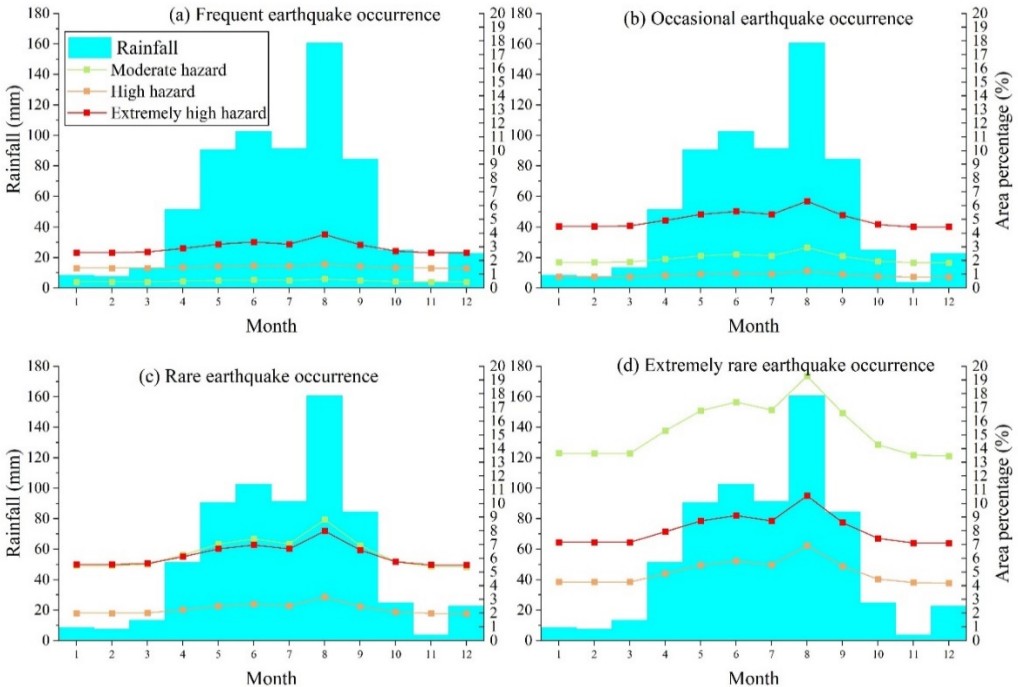

**Figure 12 The change of rainfall and PSLH with months in the condition of different earthquake occurrence scenarios.**

## 5 Conclusions and discussions

The method presented in this study, considering the uncertain features of both earthquakes and rainfalls, is effective in
assessing the PSLH in earthquake-prone mountainous areas suffering from heavy rainfalls. The assessment of future PSLH
in Bomi, China shows that the spatio-temporal variability of rainfall influences significantly the pattern of potential seismic
landslide hazards. In the frequent or occasional earthquake scenarios, the majority area of Bomi is deemed to have a low
hazard level, with only 668.25 and 875.92 km$^2$ areas (accounting for 3.99% and 5.23% of Bomi) being categorized as risky
zones (high and extremely high Hazard levels). However, in the rare or extremely rare earthquake scenarios, there is a
notable increase in areas being classified as risky zones, with 1246.1 and 1884.2 km$^2$ areas (accounting for 7.44% and 11.25%
of Bomi) being assessed as extremely high hazards. The high and extremely high hazard areas are situated mainly in the
southern part of Bomi and along the Parlung Zangbo River, where the targeted hazard mitigation measures are very required.
Notably, future seismic events are expected to interact synergistically with heavy rainfalls in the rainy season, especially in
August, which will enlarge the landslide hazard area and raise the hazard level. The findings in this study are instructive for



the planning and implementation of landslide prevention measures in Bomi, guiding the stakeholders and local authorities to prioritize infrastructures, settlements, and interventions for possible increasing vulnerability in the future.

The acquisition of seismic parameters under future earthquake scenarios serves as an important reference for the zoning of
seismic landslide hazards. This study adopts the fifth seismic ground motion parameters zonation map of China, which provides a probabilistic estimate of the potential earthquake impacts. While the zonation map is scientifically rigorous, its estimates of earthquake impacts on PSLH are often conservative. Seismic parameters recorded in historical earthquakes indicate that actual observed parameters are significantly higher than those corresponding to the scenarios depicted in the zonation map. Therefore, under actual earthquake scenarios, the risk of seismic landslides will be even more severe than that
assessed based on the zonation map. Furthermore, the low-value regions indicated in the zonation map do not mean being immune to the disturbance of strong earthquakes. However, in extreme earthquake scenarios, these areas might experience also severe seismic landslide disasters than anticipated.

Rainfall is also a crucial factor affecting slope stability, and the dynamic distribution of rainfall further influences the spatio-temporal distribution of seismic landslide hazards. This study comprehensively considers rainfall information across
different months based on historical rainfall data to assess the landslide hazard under different earthquake scenarios, and thereby quantify the spatio-temporal distribution of potential seismic landslide hazard under the combined effects of earthquakes and rainfalls. Although the rainfall information based on historical rainfall data can offer insights into the spatio-temporal distribution of rainfall, future rainfall conditions cannot be represented effectively, particularly in the background of global climate change where extreme rainfall events are on the rise in the future. Consequently, relying solely on
historical rainfall data to assess landslide hazards under future earthquake scenarios may pose certain limitations. It is necessary to take into account the future trends of rainfall development to assess more accurately the distribution of landslide hazards under future climate and earthquake scenarios.

The soil slopes are more significantly affected by rainfall than the rocky slopes. To distinguish the types of slopes, this study extracted soil slopes by using vegetation information as an important reference based on the landuse/landcover data.
Although the landuse/landcover data is an important indicator to distinguish between soil and rocky slopes, it still has some limitations. For example, the vegetation in the landuse/landcover data only corresponds to the area covered by vegetation canopy and does not represent its stand conditions. Besides, the accuracy and timeliness of the landuse/landcover data may further affect the results of the extracted soil slopes. Therefore, it is necessary to combine more data sources to effectively differentiate between soil and rocky slopes.
Topographic data is an important factor in regional seismic landslide hazard assessment, which directly affects the fineness and accuracy. This study used 30 m DEM data to conduct PSLH assessment under rainfall and future earthquake scenarios. Although 30 m DEM data can reflect the spatial distribution of seismic landslide hazards at a certain scale, it may not be able to capture the subtle changes or details in localized topography, which often has important impacts on the occurrence of seismic landslides. High resolution DEM data should be used in the PSLH assessment to obtain finer results and provide
better support for disaster prevention and mitigation work.





**Data availability**

Data support the research obtainable from the corresponding author upon reasonable request.

**Authorship contribution**

Conceptualization of this study, structural design and academic/linguistic revision of the paper, Wu Lixin; experiment design
and original draft writing, Chen Shuai; experiment discussion and analysis, Miao Zelang. All authors have read and agreed
to the published version of the manuscript.

**Competing Interests**

No potential conflict of interest was reported by the authors.

**Acknowledgments**

The research is jointly supported by the National Key Research-and-Development Project (2023YFE0208000), and the
National Science Funds of China (No.42071256, No. 42171084).

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
