# Peer review of "Probabilistic Seismic Landslide Hazard Assessment Considering Different Scenarios of Earthquake and Rainfalls in Bomi, China"

_EGUsphere, 2024_

## Author Comment (AC2)

Referee(s)' Comments to Author:
Referee: 2

Comments to the Author
The article "Probabilistic seismic landslide hazard assessment considering different scenarios of earthquake and rainfalls in Bomi, China" suggests the use of a new approach for the estimation of landslide hazard over large areas. This approach is a combination of a probabilistic seismic landslide hazard (PSLH) assessment with a rainfall infiltration model which accounts for the variation of soil slope saturation conditions through the calculation of the wetting front depth. In this way, it should be possible to obtain several indications about the proneness to slope failure in response to a combined action of earthquakes and rainfalls.

Although the concept of the manuscript is promising, I believe that it cannot be considered for publication, at least at this stage. To be honest, I have serious concerns about the validity of the methodological approach the authors propose. In particular, even if I may agree with the PSLH analysis (which, however, is not a novelty for your study area, as testified by the articles the authors cited, e.g. Du et al. 2022), I believe that the infiltration model has various critical aspects. These issues mainly concern the excessive simplification of the input parameters.

**Response:** We thank sincerely the reviewer for the valuable time on the reviewing and the professional expertise in landslide study. We have thoroughly examined, modified and revised the manuscript according to your comments and suggestions. Responses are listed as below.

Comment 1:
- Line 122: "Previous studies suggested that the distribution of landslides is highly related to monthly rainfalls (Mathew et al., 2014; Tang et al., 2017). Therefore, we selected the monthly cumulative rainfall intensity in this study as the input of the Green-Ampt model". Fine, but the parameter the authors use in their model is soil saturation (as parameter m): which is the link between rainfall amount/duration and soil saturation? If the authors want to use rainfall as primary input parameter, it is fundamental to investigate the hydraulic response of the soil to different types of expected rainfalls. For substantiating their strong choice of monthly rainfall, the authors cannot simply rely on the above-mentioned three lines…. Otherwise, it makes more sense to carry out a parametric analysis of soil saturation.

**Response:** Thanks. In this study, soil saturation is defined as the percent of the saturated portion of a potential landslide, i.e., the ratio of the depth of saturated soil ($z_w$) to the potential landslide thickness ($z$). To quantify the depth of saturated soil, we introduced a classical hydrological model (i.e., Green-Ampt model) to simulate approximately the rainfall infiltration processes under different rainfall conditions. The rainfall condition and the hydraulic response of soil to different rainfalls are fundamental to accurately simulate rainfall infiltration. However, such information is difficult to obtain, especially at a regional scale. Although the the hydraulic response of soil to different rainfalls is critical in the hydrological model, which will affect notably the depth of saturated soil, the impact of the hydraulic response of soil to rainfall was not considered in this study since the chief purpose of this manuscript focused on the impact of different rainfall conditions on the calculation of saturated soil depth. Besides, the rainfall data used in this study were derived from a reanalyzed product with a temporal resolution of one month, which is insufficient to support a parametric analysis to explore the impact of different rainfall durations on the probabilistic seismic

landslide hazard assessment. Therefore, we selected the monthly cumulative rainfall as the rainfall duration in the Green-Ampt model in this study. Once the rainfall data with higher temporal resolution are available, the effects of different rainfall durations on probabilistic seismic landslide hazard assessment will be further investigated in the future.

Comment 2:

- Line 169: "Physical properties, such as the saturated water content, the initial water content, and the saturated hydraulic conductivity are also important in landslide hazard assessment. Since these parameters are not yet available at a regional scale, we assume that all the Groups in Bomi have the same saturated water content, the initial water content, and the saturated hydraulic conductivity. According to the previous studies in Bomi, the saturated water content of the slopes is usually in the range of 30-50%, and the initial water content is in the range of 10-30%(Liu et al., 2020; SU and LI, 2020). We adopt the average values in previous studies as the saturated water content and initial content of the slopes, being 40% and 20%, respectively." This is another huge simplification, especially in relation to the (extremely) large scale of the study area. Different lithologies imply different hydraulic properties, and the authors should put strong efforts in order to attribute reasonable values to each material. Otherwise, the obtained results could be scarcely significant.

**Response:** Thanks a lot. Different lithologies indeed imply different hydraulic properties. However, obtaining detailed and accurate hydraulic properties parameters at a regional scale is challenging, especially in this plateau area rarely investigated due to inconvenient transportation. Existing studies usually assign empirical hydraulic properties parameters to different lithologies for regional scale analysis. Since the Green-Ampt model selected in this manuscript is applicable to the soil slopes, only the soil slopes (i.e., the area covered by vegetation) were conducted for rainfall infiltration studies. These soil slopes typically have similar lithologic and hydraulic properties. We, therefore, adopted the same hydraulic property parameters without considering the different hydraulic properties of different lithologies. It is worth noting that detailed and accurate hydraulic property parameters will help to further improve the reliability of the rainfall infiltration simulation, and in the absence of such data, appropriate parameter simplifications are still acceptable. Once detailed and reliable data information is available, we will conduct a finer-scale probabilistic seismic landslide hazard assessment in the future.

Comment 3:

- Line 262: "To simplify the operation and reduce the input parameters, we assumed that the saturated hydraulic conductivity is greater than the rainfall in Bomi, and thus the wetting front depth can be calculated with the Green-Ampt model." I think that this sentence summarizes and confirms what I said in the preceding two points….

**Response:** Thanks very much for mentioning this. Although we have conducted two field investigations in Bomi in 2020 and 2021, respectively, the detailed hydrogeological information about this study area is still insufficient. To this end, according to the information collected from the field investigations, the necessary simplifications and assumptions were made on the hydrological parameters at the regional scale, including ignoring the differences in the hydraulic properties of the different lithologies and assuming that all rainfall infiltrates into the soil, i.e., the saturated hydraulic conductivity is greater than the rainfall intensity. Although such assumptions and simplifications

will further affect the reliability of the regional seismic landslide hazard assessment to a certain extent, it is acceptable as compared to that ignoring the rainfall effect on regional seismic landslide hazard assessment.

Comment 4:

- Beyond the infiltration model, it is possible to notice a certain vagueness and uncertainty also in relation to other types of data. For instance, in line 223 the authors assert that: "Since the global empirical depth parameter of the potential landslide was usually used for the CRMSH-PD model in previous studies (Shinoda and Miyata, 2017; Zang et al., 2020), we set the depth of potential landslides as 3 m referring to the depth of landslides from the historic landslide inventory in Bomi". Actually, in the cited papers the investigated landslides have a depth "less than 3 m", which is not "an average value of 3 m".

**Response:** Thanks for mentioning this point. For the information on the depth of the potential landslide, we assumed that the depth of the landslide is constant at the regional scale, and adopted an empirical parameter (with a maximum of 3m) referring to the relevant literature information. In the revised manuscript, we have added the references for the depth of the potential landslide: "Since the global empirical depth parameter of the potential landslide was usually used for the CRMSH-PD model in previous studies (Shinoda and Miyata, 2017; Zang et al., 2020), we set a maximal and constant depth of potential landslides as 3 m referring to the previous study (Du et al., 2017)".

Comment 5:

- Furthermore, there is no information about these historical landslides in Bomi. The authors mention different times this inventory throughout the manuscript, but they do not provide any detail in this sense (e.g., how many landslides are included? Is it a multi-temporal or an event-based inventory? Which types of landslides are included? Rainfall-induced? Earthquake-induced? Both?). Again, all these approximations and lack of information do not help in substantiating the reliability of your results. A possible solution could come from a strong validation process. But, unfortunately, the manuscript is quite poor also from this point of view. For instance, in line 225 the authors assert that: "According to the yield acceleration calculated with the CRMSH-PD model, the study area is divided into five susceptibility levels. Higher susceptibility levels correspond to lower slope stability, and more historical landslides developed. The results show that the history landslides are distributed mainly in areas with high and extremely high susceptibility levels, accounting for 76.3% of the total landslides, and the number of landslides increases significantly with the landslide susceptibility level." In the light of the lack of information that I mentioned before, it is impossible for the reader to verify what the authors are saying. Are the obtained results reasonable? Maybe, but in the absence of a real validation, Figure 9 and the following ones just represent a useless exercise

**Response:** Thanks for your kind comments. In the revised manuscript, we add more information about the inventory as: "*In addition, the history hazard inventory was also used in this study. The inventory data was provided by the China Geological Survey (CGS), which contains all geologic hazards that occurred in Bomi County during the period 2000-2019. The inventory recorded a total of 140 geologic hazard sites, including 28 landslides and 112 debris flows*". In this study, to verify the validity of the results, verification was also performed for the critical input parameter (critical acceleration) in the CRMSH-PD model. In the revised manuscript, we have added detailed

descriptions as: "*The yield acceleration is crucial for subsequent PSLH assessment. Studies have shown that the yield acceleration can effectively reflect the slope stability in non-seismic scenarios. Therefore, the distribution of historical hazards can be used to verify the effectiveness of the calculated yield acceleration. This study divided Bomi County into four susceptibility levels based on the calculated yield acceleration, and the effectiveness of the susceptibility zoning was verified using historical hazard inventories. The results show that most of the hazards are distributed in high-susceptibility areas, and the number of hazards increases as the susceptibility level increases. As a result, the critical acceleration calculated with the CRMSH-PD model can reflect slope stability in non-seismic scenarios to a certain extent, which is suitable for the subsequent PSLH assessment.*" For the probabilistic seismic landslide hazard assessment under future earthquake scenarios, the validity of the results is to be evaluated through real scenarios in the future. Whether the results are consistent with existing empirical knowledge is a possible way to evaluate the validity of the assessment results. In the revised manuscript, we added more description about the assessment results as: "*This paper considers the probabilistic seismic landslide hazard assessment under the combined influence of earthquake and rainfall. The assessment results reflect not only the influence of different rainfall on the distribution of seismic landslide hazards but also the influence of different seismic scenarios on the distribution of seismic landslide hazards. Compared to the assessment results from independent consideration of the triggering factors such as rainfall or earthquakes, these results from combined consideration could be more in line with the landslide hazard under real scenarios, especially for the earthquake-prone mountainous areas suffering from heavy rainfalls.*"

Comments 6:

- "Introduction": this section should be improved with more international references, mentioning alternative approaches for landslide hazard assessment due to the combined action of earthquakes and rainfalls. This is a research topic that various authors have already dealt with.

**Response:** Thanks for your kind suggestion. In the revised manuscript, we have cited more international references relevant to the field. Besides, the research questions and objectives have been reformulated in the revised introduction.

Comment 7:

- Figure 2 is very chaotic and does not clarify the different steps of the proposed approach. Please modify it in order to be coherent with the text.

**Response:** Thank you very much. In the revised manuscript, we have redrawn the Fig.2 as in the following:

[Figure]

*Fig.3 Flowchart of the novel method for PSLH assessment considering both earthquakes and rainfalls.*

Comment 8:
- Line 147-148: "As a result, the scale and frequency of geological hazards in Bomi increased significantly compared to that before the great earthquake". Please clarify this sentence. Are you talking about post-seismic landslides? (sensu Fan et al., 2019)?.

**Response:** Yes, and thanks. In response to the suggestions of the other reviewer, we have reorganized the section in the revised manuscript as: *"Bomi County, located in the southeast of the Tibetan, is a region with the characteristics of fragmented geological structure and complicated topography and geomorphology. The region is a geo-hazards-prone region of high seismic activity, and several strong earthquakes have occurred frequently because of the regional strong tectonic movements(Zhao et al., 2023). According to the earthquake data of the U.S. Geological Survey (USGS), from January 1, 1990, to January 1, 2024, a total of 439 earthquakes have been recorded in Bomi County and its nearby (within a 100km buffer zone), as shown in Fig. 1. For example, the M8.6 Motuo earthquake in 1950 was one of the strongest earthquakes since 1850, which induced a large number of collapses, landslides, and other geologic hazards that caused a large number of casualties(Li et al., 2021). In addition, the rainfall pattern in Bomi County varies significantly in both spatial and temporal domains, with the rainfall mainly concentrated in June, July, and August. Bomi County also plays a pivotal role in the overall economic and social development of southeast Tibet. Several major infrastructures (such as the Sichuan-Tibet Railway) and urbanization engineering have been planned in this region. Threatened by the combined effects of earthquake and rainfall, the settlements, infrastructures, and engineering will be exposed to a huge risk of earthquake and seismic landslide throughout the entire cycle of construction and operation. To effectively avoid, control, and reduce the risk of seismic landslides, assessing the spatial distribution of seismic landslide hazards across various future*

*earthquake scenarios and different rainfall conditions is of practical significance.”*

Comment 9:

- Figure 5, 6 and 7: please add the monitoring stations. It would be also interesting to understand how authors spatialized the punctual data.

**Response:** Sorry for making the confusion. The regional rainfall data used in this study were not derived from the monitoring stations, but rather from the reanalyzed rainfall data products publicly released. In the revised manuscript, we add more information about the resource of the rainfall data as: *“The rainfall data are derived from the 1-km resolution month-by-month precipitation dataset of China (1901-2021) (Peng, 2020). The dataset was generated with the Delta spatial downscaling scheme based on the global 0.5° climate dataset released by CRU and the global high-resolution climate dataset released by WorldClim. The rainfall data was validated using data from 496 independent meteorological observation stations, including the Bomi station, and the results were verified to be credible.”*

Comment 10:

- Figure 8-11: in my opinion, these figures are far from explanatory, since it is very difficult to distinguish different susceptibility levels. Please modify them.

**Response:** Thanks a lot. In the revised manuscript, we have used different colormaps to distinguish different landslide susceptibility levels. The modified figures are as follows:

[Figure]

*Figure 8 Maps of seismic landslide hazard levels in conditions of frequent (a), occasional (b), rare (c), and extremely rare (d) earthquake occurrence scenarios.*

[Figure]

*Figure 9 Changes of PSLH level in the condition of four earthquake occurrence scenarios: (a) Fig. 8a minus Fig. 8b, (b) Fig. 8c minus Fig. 8b, (c) Fig. 8d minus Fig. 8b.*

[Figure]

*Figure 10 PSLH maps considering the spatial variability of rainfall in conditions of frequent (a), occasional (b), rare (c), and extremely rare (d) earthquake occurrence scenarios.*

[Figure]

*Figure 11    PSLH maps in different months referring to different earthquake occurrence scenarios.*

Once again, we appreciate your carefully reading on the manuscript and your kindness on helping improve the quality. The comments and suggestions are very valuable for the revision to improve the quality and readability of the paper.

Best regards,
Lixin and Shuai

---

## Author Comment (AC3)

Referee(s)' Comments to Author:
Referee: 1

Comments to the Author
- Review for "Probabilistic Seismic Landslide Hazard Assessment Considering Different Scenarios of Earthquake and Rainfalls in Bomi, China" by Chen Shuai et al.
- Landslides are a common phenomenon in mountainous regions. Among the most common triggers of landslides are rainfall and earthquakes. In this manuscript the authors follow an approach that includes rainfall in a probabilistic seismic hazard assessment workflow.
- The landslide hazard is categorized based on the permanent seismic displacement, which is a function of the yield acceleration and the peak ground acceleration. In the calculation of the yield acceleration they account for spatial heterogeneity of the rock mass strength by using a scaling factor for the initial yield acceleration. The latter is dependent among others on the water saturation of the subsurface. This in turn is computed with the wetting front depth, which itself is dependent on the rainfall intensity.
- They apply their model to Bomi County at the eastern Himalayan syntaxes, a region of strong seismicity and landslide hazard. They compare models that account for rainfall with those that ignore rainfall. They find that the spatiotemporal distribution of rainfall has a substantial impact on the landslide hazard assessment.
- Although I see the usefulness of such a model, I have doubts that the used data is sufficient for the task. One striking example is the rainfall data, which relies on a single station. There is no information how the rainfall for the entire county has been derived from it. In addition, how were the mechanical properties determined? There is too few information on the landslide inventory as well. How was the performance of the model evaluated?
- I also suggest restructuring the introduction to also include the study area. Furthermore the research question should be clearly formulated in the introduction.

**Response:** We thank the reviewer for the time and expertise they have invested in these reviews. We'll reply to individual points below.

Comment 1:
- Title: In the title you should write "Bomi County" instead of just "Bomi". After all you did the assessment for the entire county not just for the town.

**Response:** Thanks. In the revised manuscript, the title has been rewritten as: " Probabilistic Seismic Landslide Hazard Assessment Considering Different Scenarios of Earthquakes and Rainfalls in Bomi County, China".

Comment 2:
- L30: Please clarify whether these people were all victims to landslides.

**Response:** Thanks. To avoid misunderstandings, in the revised manuscript, we have reorganized the first paragraph as: "Seismic landslides are natural hazard phenomena of slope stability-lossing induced directly or indirectly by earthquakes, and a strong earthquake can usually induce a large number of seismic landslides in a short period. For example, the 2008 Mw 7.9 earthquake that occurred in Wenchuan induced more than 197,000 seismic landslides immediately (Cui et al., 2011; Xu et al., 2014). As a common geological disaster of earthquakes, seismic landslides usually result in significant

casualties, infrastructure damage, and substantial property losses, which is a major threat in earthquake-affected areas (Fan et al., 2019; Tang et al., 2017). Probabilistic Seismic landslide hazard (PSLH) assessment aims at predicting and assessing the distribution of potential seismic landslide hazards that may be triggered by future earthquake scenarios, which is an important way to mitigate the impact of seismic landslides. The results of the PSLH assessment can provide important references for medium- and long-term landuse planning of urbanization and engineering site selection of major infrastructures(Gerstenberger et al., 2020; Rollo and Rampello, 2021; Wang and Rathje, 2015)".

Comment 3:
• L55: "few studies have considered…" references of a few studies needed
**Response:** Thanks. In the revised manuscript, the references have been added.

Comments 4:
• L66-67: "…in recent years (JIbson, 1993; Wieczorek et al., 1985) …" if it has become widely used in recent years why not citing more recent studies?
**Response:** Thanks. In the revised manuscript, we have cited more recent studies.

Comment 5:
• L72: What does CRMSH stand for.
**Response:** Sorry for this. The CRMSH stands for Considering the Rock Mass Spatial Heterogeneity (CRMSH), which has been added in the revised manuscript.

Comment 6:
• L73: "… 2008 Wenchuan M7.9 earthquake" a reference needed.
**Response:** Thanks. These two earthquake cases came from the same reference.

Comment 7:
• L74: ".. that the geo-environment characteristics of Bomi are much similar to that of Wenchuan and Ludian" Could you justify this claim somehow? Both, Ludian and Wenchuan are ~800 km away from Bomi.
**Response:** Thanks. The method we refer to quantifies the spatial heterogeneity of the rock mass by primarily considering the effects of topography and tectonic setting on the strength of the rock mass. The topography and tectonic setting in Bomi County are also complex, and thus we suggest that the effects of topography and tectonic setting on the strength of the rock mass need to be considered when quantifying the strength of rock mass. To avoid misunderstanding, in the revised manuscript, we rewrote the sentence as: "In the CRMSH-PD model, topography and tectonic setting are the two main factors used to quantify the spatial heterogeneity of regional rock mass. Since Bomi County has complex topography and tectonic settings, this paper adopts the CRMSH-PD model for PSLH assessment in Bomi County to further consider the spatial heterogeneity of regional rock mass."

Comment 8:
• L95: This should be the first equation because it is the metric you use to define the landslide hazard, right?
**Response:** Thanks for your concern and question. In the revised manuscript, we have reorganized

**Comment 9:**

- L116-123: These lines and equations can be removed because they are not used after all (see comment below).

**Response:** Nice suggestion, thanks. Equations in L116-123 have been removed in the revised manuscript.

**Comment 10:**

- L124: So you converted the monthly value (mm/month) into m/s?

**Response:** Yes. In the manuscript, we use monthly rainfall intensity (mm/month) to calculate the depth of the wetting front in the GA model, since the rainfall data we obtained is monthly cumulative rainfall data.

**Comment 11:**

- L131: the variable t has already been defined as rainfall duration in L111.

**Response:** We feel sorry for this. The error has been corrected in the revised manuscript.

**Comment 12:**

- L138: Study area should go to the introduction.

**Response:** Thanks for your nice suggestion, the study area has been moved to the introduction.

**Comment 13:**

- L148-151: "For example …" It is not clear how this relates to the great earthquake in the previous sentence.

**Response:** Sorry for this. Previous studies have revealed that the Yigong landslide was strongly affected by the great Motuo earthquake. In the revised manuscript, these sentences have been removed.

**Comment 14:**

- L162 "the slopes in Bomi were divided into five different groups, i.e., Groups 1~5" this is confusing, you divide the lithologies not the slopes.

**Response:** Thanks for pointing out this expression wrong. In the revised manuscript, the sentence has been rewritten as: "According to the 1:250,000 engineering geological map and previous studies(Chen et al., 2019; Ma and Xu, 2019), the lithologies in Bomi County were divided into five different groups, i.e., Groups 1~5."

**Comment 15:**

- L165: "The mechanical parameters … are then assigned referring to previous studies (Du et al., 2022)." First of all you write "studies" but give only a single study for reference. Secondly, I checked the study and they used different values for their lithologies, which are also not exactly the same as they occur in Bomi. For example, for loose sediments they use c' = 25, phi'=24 and gamma=14, whereas you use 17-23, 22-28 and 10-14, respectively. Similar deviations can be observed for the other lithologies as well. How do you come up with these

numbers? In general, there is a lack of information how the mechanical properties have been determined.

**Response:** Thanks. The mechanical parameters of the rock mass were assigned referring to the data given in the reference and the information gathered by the field geological surveys of Bomi County in 2020 and 2022. It is worth noting that the yield acceleration calculated based on the mechanical parameters of the rock mass given in the references is generally high, and the distribution of the calculated yield acceleration is poorly matched with the distribution of historical geologic hazards, i.e., there is an overestimation of the yield acceleration in Bomi County. According to the field geological surveys, the rock mass in Bomi County is much more fractured and the weathering is more serious, so we suggest that the mechanical parameters of the rock mass in Bomi County should be lower than that given in the literature. Besides, considering the inherent uncertainty and data difference of the mechanical parameters of the rock mass, we assigned parameter intervals for different groups to reflect the uncertainty of the mechanical parameters of the rock mass. In the revised manuscript, the sentence has been rewritten as: "The empirical mechanical parameters (unit weight, effective friction angle, and effective cohesion) of these groups are then assigned referring to the previous study(Du et al., 2022) and field geological surveys. Besides, considering the inherent uncertainty and data difference of the mechanical parameters of the rock mass, we assigned parameter intervals for different groups to reflect the uncertainty of the mechanical parameters of the rock mass, as shown in Table 1. "

Comment 16:
- L165: "different groups" which ones? Be specific.

**Response:** Sorry. In the revised manuscript, the sentence has been rewritten as: "In this study, the median values were used as the representative mechanical parameters in the CRMSH-PD model according to the parameter intervals of the assigned different groups."

Comment 17:
- L170-172: "Since these parameter …" This is a strong assumption that needs to be discussed.

**Response:** Thanks for raising this point. In the revised manuscript, the sentence has been rewritten as: "These parameters are critical in the GA model, while not yet available at a regional scale. Since this study is primarily oriented toward the assessment of potential seismic landslide hazards at the regional scale, the necessary parameter simplifications are acceptable. Therefore, we made reasonably a general assumption that all the classified Groups in Bomi have the same saturated water content, the initial water content, and the saturated hydraulic conductivity."

Comment 18:
- L175: Table caption: It would help the reader to again write out the names of these parameters here. Further, start description of the groups with group one on the top.

**Response:** Thanks for your kind suggestion. The manuscript has been revised accordingly.

Comment 19:
- L177: What radius was used for calculating the relief?

**Response:** The topography relief was derived with the Block Statistics module in ArcGIS with a circle radius of 5 pixels.

Comment 20:

• L178: What is tectonic distribution? Do you mean the distribution of faults?

**Response:** Sorry for making this confusion. We did use the distribution of faults to reflect the tectonic distribution since the fault is an important component of the tectonic distribution. In the revised manuscript, we added more detailed information about the distribution of faults: "To reflect the tectonic setting influence on seismic landslide, the 1: 2,500,000 fault distribution map was used, and the distance to the fault was extracted using the Euclidean Distance module in ArcGIS. The distance to the fault was further divided into 10 classes, as shown in Fig. 4e."

Comment 21:

• L181: "1:1,000,000 river distribution" reference needed; "using ArcGIS" What exactly did you do?

**Response:** Thanks. In the revised manuscript, we have added more detail about the river distribution: "Besides, the 1: 1,000,000 river distribution map, which can be downloaded from the National Catalogue Service For Geographic Information (https://www.webmap.cn/commres.do?method=result100W), was used to reflect the hydrography setting influence on seismic landslide. The distance to the river was derived using the Euclidean Distance module embodied in ArcGIS, and was further divided into 7 classes, as shown in Fig. 4f."

Comment 22:

• L187: As stated above there is way too few information regarding this inventory. Reference? How was it created? Looks like the landslides cluster along roads. Is there perhaps an observational bias? How many landslides are there? Which timeframe is considered? How many were triggered by earthquakes, how many by rainfall, and which ones?

**Response:** Thanks for the questions. In the revised manuscript, we add more information about the inventory: "In addition, the history hazard inventory was also used in this study. The inventory data was provided by the China Geological Survey (CGS), which contains all geologic hazards that occurred in Bomi County during the period 2000-2019. The inventory recorded a total of 140 geologic hazard sites, including 28 landslides and 112 debris flows."

Comment 23:

• 192: How was the rainfall distribution derived from only a single station within the county? There is also a reference needed.

**Response:** We feel sorry for this. In the revised manuscript, we add more information about the resource of the rainfall data: "The rainfall data are derived from the 1-km resolution month-by-month precipitation dataset of China (1901-2021) released by Peng (Peng, 2020). The dataset was generated by the Delta spatial downscaling scheme at the regional downscaling in China based on the global 0.5° climate dataset released by CRU and the global high-resolution climate dataset released by WorldClim. Moreover, the rainfall data was validated using data from 496 independent meteorological observation stations, including the Bomi station, and the results were verified to be credible."

Comment 24:

- L206-209: Can you give the moment magnitudes that correspond to these scenarios?

**Response:** In the fifth ground shaking parameter zoning map of China (GB18306-2015), the differences between these scenarios are only reflected in the seismic motion parameters, and the scenarios do not correspond to specific moment magnitudes.

Comment 25:
- L210: Who created the PGA map? If you did it, how? It is not mentioned in the methodology. And then it should go to the results.

**Response:** In the revised manuscript, we have added more detail about the PGA map. "The ground shaking parameters used in the study were derived from the ground shaking parameter zoning map of China (GB18306-2015) (General Administration of Quality Supervision et al., 2015). The zoning map provides peak ground acceleration at different levels of exceedance probability in different areas based on the probabilistic seismic hazard analysis method."

Comment 26:
- L220: Table 2: You list only 12 scenarios with the monthly rainfall intensities but later you present 48 scenarios in Fig.11.

**Response:** We feel sorry for this. In this study, a total of 56 different scenarios of probabilistic seismic landslide hazard mapping were carried out, including 4 without rainfall and 52 with rainfall. In the new manuscript, we have revised the errors in Table 2.

Comment 27:
- L226: "Higher … " How did you incorporate historic landslides in the calculation?

**Response:** In this manuscript, the historic hazard inventory was used to validate the reliability of the yield acceleration calculated based on the CRMSH-PD, since the mechanical parameters of rock mass in the CRMSH-PD model are assigned based on empirical, which has great uncertainty. In the revised manuscript, we have added detailed descriptions as: "The yield acceleration is crucial for subsequent PSLH assessment. Studies have shown that the yield acceleration can effectively reflect the slope stability in non-seismic scenarios. Therefore, the distribution of historical hazards can be used to verify the effectiveness of the calculated yield acceleration. This study divided Bomi County into four susceptibility levels based on the calculated yield acceleration, and the effectiveness of the susceptibility zoning is verified using historical hazard inventories. The results show that most of the hazards are distributed in high susceptibility areas, and the number of hazards increases as the susceptibility level increases. As a result, the critical acceleration calculated with the CRMSH-PD model can reflect slope stability in non-seismic scenarios to a certain extent, which is suitable for the subsequent PSLH assessment."

Comment 28:
- L238-243: How do you explain the spatial distribution of hazard zones?

**Response:** Thanks for your question. In the revised manuscript, we have added more detailed information to explain the spatial distribution of hazard zones as: "Fig. 8 shows the distribution of seismic landslide hazards in the condition of four potential earthquake scenarios. Ignoring the influence of rainfall, the distribution of seismic landslide hazard is mainly affected by the seismic factors, and the high-risk zone of probabilistic seismic landslide is mainly distributed in the region

of high value of seismic parameters, i.e., high peak ground acceleration. In addition, for different future earthquake scenarios, there are significant differences in the area of seismic landslide hazard zones. Specifically, for the scenarios of frequent and occasional earthquake occurrence, only a small area is assessed as high hazard level, which distributes mainly in the south of Bomi; for the scenarios of rare and extremely rare earthquake occurrence, the area of high and extremely high levels increases significantly, which scatters around the Bomi town and along the Parlung Zangbo River."

Comment 29:
• L244: Where are these statistics shown?

**Response:** Sorry for this. In the new manuscript, we have revised the sentence L244 as:" To quantify the differences in seismic landslide hazard under different earthquake scenarios, we computed the area corresponding to different hazard levels. The results show that the areas of moderate, high, and extremely high PSLH levels increase significantly with the earthquake scenario changing from frequent occurrence to extremely rare occurrence."

Comment 30:
• L262-263: Here you state that you assume that the saturated hydraulic conductivity is always greater than the rainfall in Bomi. So you only use equation 7, right? Then I suggest removing equations 8 and 9. What value do you assign to t in equation 7?

**Response:** Thanks for your nice suggestion. In the new manuscript, the equations 8 and 9 have been removed from the manuscript. In addition, previous studies have shown that the distribution of landslides is correlated with monthly cumulative rainfall, and this study also uses monthly cumulative rainfall intensity.

Comment 31:
• L292: "is effective in assessing the PSLH in earthquake-prone mountainous" How did you evaluate the effectiveness of your model?

**Response:** Thanks for your question. For probabilistic seismic landslide hazard assessment under future earthquake scenarios, it is difficult to evaluate the validity of the results through real data. Whether the results are consistent with existing empirical knowledge is a possible way to evaluate the validity of the assessment results. In the new manuscript, the sentence has been reorganized: "This paper considers the probabilistic seismic landslide hazard assessment under the combined influence of earthquake and rainfall. The assessment results can not only reflect the influence of different rainfall on the distribution of seismic landslide hazards but also the influence of different seismic scenarios on the distribution of seismic landslide hazards. Compared to the assessment results of landslide hazard under independent consideration of the triggering factors such as rainfall or earthquakes, this combined results are more in line with the landslide hazard under real scenarios, especially for the earthquake-prone mountainous areas suffering from heavy rainfalls."

Comment 32:
• L293: "significantly" Can you show that it is significant in a strictly statistical sense?

**Response:** Thanks. To avoid misunderstandings for readers, the "significantly" was removed from the sentence in the revised manuscript.

Comment 33:

- L306: "While the zonation map … zonation map." This statement makes me question the validity of the zonation map. You refer to seismic parameters that are higher in the records than those that correspond to the scenarios in the zonation map. Can you give numbers for both? Which seismic parameters are you referring to?

**Response:** Thank you for your nice question. Zoning maps provide regional averages of ground shaking accelerations, whereas actual earthquake scenarios may produce higher or lower ground shaking accelerations than the average, so it is inaccurate to say that the seismic parameters that are higher in the records than those corresponding to the scenarios in the zonation map.

Comment 34:

- L333: "details in localized topography, which often has important impacts on the occurrence of seismic landslides" please specify?

**Response:** Thank you for your question. In the revised manuscript, we have added more detail information about the resolution of the topographic data: "The resolution of topographic data is an important factor in regional seismic landslide hazard assessment, which directly affects the fineness and accuracy. In this study, topography-related factors are extracted based on 30 m resolution DEM for PSLH assessment under rainfall and future earthquake scenarios. Although the 30 m resolution DEM data can reflect the regional distribution of topography and general features, it may not capture subtle changes or details of localized topography, which often have important implications for the occurrence of seismic landslides. For example, previous study has shown that the permanent seismic displacement decreases with decreasing DEM resolution, and the difference is mainly concentrated in the steep slope area(Wang et al., 2017). The reason is that the slope angle is an important parameter of the PD model, and the high-resolution DEM can get a more detailed and accurate slope data, while the relatively low-resolution DEM usually underestimates the local slope(Chang and Tsai, 1991). Besides, DEM resolutions can also affect the result of seismic landslide hazard assessment by affecting the evaluation of topographic effect. Post-earthquake investigation indicates that the topographic amplification effect on ground motions will aggravate earthquake-induced landslide disasters(Shafique et al., 2011). The joint use of high-resolution DEM and detailed geological and geotechnical information makes it possible to take the subsurface and topographic conditions into account to assess the correct value of the ampli fified PGA at the site and to draw reliable permanent seismic displacement maps based on PD model."

Comment 35:

- In general I advise to use different colormaps for different things. The chosen red-green colormap is not a good choice, because it excludes every person who is color blind. Moreover, in most of the result maps the colors are practically indistinguishable.

**Response:** Thank you for your kind advice. In the revised manuscript, we have redrawn the results with different colormaps.

Comment 36:

- Fig.3: What kind of faults are depicted here? What is the source of the earthquakes -> reference? What does "historic" mean actually? Are these Earthquakes that occurred during the last century?

**Response:** Thank you for the questions. In the revised manuscript, we've added more information on the earthquakes. In addition, this paper considers only the effect of fault distribution on seismic landslides, while the fault data we downloaded did not have a detailed description of the fault occurrence and type, which were not able to be specified in Fig. 1.

Comment 37:

- Fig.5: How was this spatial distribution of rainfall derived? Why does it rain more in the valleys compared to the ranges? I would assume that there is an orographic effect that would lead to the opposite pattern?

**Response:** Thank you for your nice question. This study adopted 1-km monthly precipitation dataset for China (1901-2021) to quantify the spatial and temporal distribution of rainfall, and the patterns of rainfall distribution may be related to local topographic closure effects, i.e., orographic effect.

Comment 38:

- Fig.6. You should replace the Chinese scripts with latin ones.

**Response:** Thanks. The Chinese scripts have been replaced with Latin ones in the revised manuscript.

Comment 39:

- Fig.8: Hard to differentiate the colors. Please give some information regarding the magnitudes of EQ in these scenarios.

**Response:** Thanks. In the revised manuscript, we have used different colormaps to distinguish the seismic landslide hazard level. The modified figures are as follows:

[Figure]

*Figure 8 Maps of seismic landslide hazard levels in conditions of frequent (a), occasional (b), rare (c), and extremely rare (d) earthquake occurrence scenarios.*

Comment 40:

• Fig.9: I see no change between (a) and (b).

**Response:** Sorry for this. In the revised manuscript, we used different colormaps to highlight the change in seismic landslide hazard levels in different scenarios. The modified figures are as follows:

[Figure]

*Figure 9 Changes of PSLH level in the condition of four earthquake occurrence scenarios: (a) Fig. 8a minus Fig. 8b, (b) Fig. 8c minus Fig. 8b, (c) Fig. 8d minus Fig. 8b.*

Comment 41:

- Fig.12: The way you show the hazard is confusing. I would assume that affected area decreases with the hazard level, not the other way around. Since areas with extreme hazard level are also incorporating areas with lower hazard levels, the former should always be smaller than the latter.

**Response:** Thanks very much for pointing out this. In the revised manuscript, Fig.12 has been replaced with a new picture. The new figures are as follows:

[Figure]

*Figure 12 The change of rainfall and PSLH with months in the condition of different earthquake occurrence scenarios.*

Once again, we appreciate your carefully reading on our paper very much. The comments and suggestions are very valuable for the revision to improve the quality and readability of the paper.

Best regards,

The authors